# Is Exercise the Best Medicine during a COVID-19 Pandemic? Comment on Constandt, B.; Thibaut, E.; De Bosscher, V.; Scheerder, J.; Ricour, M.; Willem, A. Exercising in Times of Lockdown: An Analysis of the Impact of COVID-19 on Levels and Patterns of Exercise among Adults in Belgium. *Int. J. Environ. Res. Public Health* 2020, *17*, 4144

**DOI:** 10.3390/ijerph17165730

**Published:** 2020-08-08

**Authors:** Tamara Hew-Butler, Valerie Smith-Hale, Matthew Van Sumeren, Jordan Sabourin, Phillip Levy

**Affiliations:** 1Kinesiology, Health, and Sport Studies, Wayne State University, Detroit, MI 48202, USA; valerie.smith2@wayne.edu (V.S.-H.); msvansum@wayne.edu (M.V.S.); jsabourin@wayne.edu (J.S.); 2Department of Emergency Medicine, Wayne State University, Detroit, MI 48202, USA; plevy@med.wayne.edu

**Keywords:** COVID-19, exercise, pandemic

## Abstract

From Constandt et al.’s survey of 13,515 Belgium respondents, regular physical activity can be successfully initiated and sustained during a lockdown, with appropriate social distancing measures. Documentation that 77% of highly active people and 58% of low active people exercised as much or more following the institution of a nationwide lockdown was impressive, given that the cases of COVID-19 were accelerating at that time. The Belgian government’s central promotion of exercise, to boost both the mental and physical health of the population, likely contributed to the health, tolerance, and ultimate success of lockdown. In this commentary, we wish to pose a follow-up query which highlights the potential detrimental effects of intense exercise (competition) performed without social distancing measures. The proposed graphical abstract elucidates these possible risks, in contrast to the favorable results outlined in Constandt et al.’s study.

The health benefits of regular physical activity are widely recognized [1]. However, the public health and safety concerns of exercising during a pandemic remain unclear [2,3]. Data collected on 24,656 Chinese adults who died during the 1998 Hong Kong influenza outbreak indicated that both sedentary behavior (no exercise) and excessive exercise (>5-days/week) enhanced mortality risk [4]. While mild to moderate exercise reduce mortality, enhanced viral exposure from “excessive” exercise may be pathogenic [4]. Pathogenic exposure to high viral loads during vigorous exercise is supported by the subsequent transmission of COVID-19 from one dance instructor to 112 individuals (primary and secondary transmission, over a period of 24 days), following a 4-h vigorous dance session including 27 other instructors [5]. Additionally, the first European case of symptomatic COVID-19 involved an otherwise healthy, competitive, 38-year-old male Italian soccer player who required intensive care but subsequently recovered [6].

Thus, exercise safety in the context of an active COVID-19 pandemic likely depends on the degree of environmental exposure. Modest exercise, performed with appropriate social distancing, face coverings, hand sanitizing, and disinfection strategies, should remain widely promoted by public health officials [1]. However, vigorous training and competition, especially within close quarters, should proceed with caution, due to the enhanced exposure risks associated with the following factors: increased person-to-person and surface contact, increased respiration rates from high intensity exercise, and decreased immune function associated with maximal training efforts.

Our graphical abstract consolidates possible multiple environmental exposure risks, when an undetected asymptomatic athlete (i.e., not identified through routine screenings) participates in vigorous exercise [5] and competition. Data suggests that 74% of COVID-19 cases may be asymptomatic [7], with viral loads equivalent to those of symptomatic COVID-19 patients [8]. Virus transmission between athletes are enhanced up to 20-fold, with increases in respiratory rates [9]. More forceful exhalation augments the aerosolization of virus particles, which typically remain airborne for 3 h [10] and can travel upwards of 7–8 m [11] with normal speech [12] and respiration rates. Although the presence of SARS-CoV-2 is confirmed in saliva [13], the contamination of sweat through contact with respiratory droplets is probable, and further enhances transmission between athletes. Additional contamination of sports surfaces, clothing, and equipment is possible, since active virus particles can survive on porous and non-porous materials between 2–24 h [10], especially when decontamination is not possible during active competition. Lastly, vigorous training and competition has been shown to reduce athlete immunity, particularly in response to upper respiratory tract infections, which may further increase an athlete’s susceptibility to infection [14]. Although younger individuals are at reduced risk of morbidity and mortality associated with COVID-19 infection, they represent the highest percentage of asymptomatic carriers [15] and potential for thrombotic events [16]. The recent identification of asymptomatic SARS-CoV-2 positive athletes upon return to training underscores the potential for super-spreader events within entire sports teams [17].

In summary, healthy amounts of regular moderate physical activity can be safely maintained (or increased) with coordinated, consistent, and centralized public health guidance [1]. However, we caution against extending these health benefits of (socially distanced) exercise to organized competitive sport (non-socially distanced), whereas the threat of virus transmission likely undermines the positive mental and physical health benefits. Further prospective studies are required to enhance these survey data, to enhance our understanding of safe versus unsafe exercise in relationship to sport.

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
