# Peer review of "Is Exercise the Best Medicine during a COVID-19 Pandemic? Comment on Constandt, B.; Thibaut, E.; De Bosscher, V.; Scheerder, J.; Ricour, M.; Willem, A. Exercising in Times of Lockdown: An Analysis of the Impact of COVID-19 on Levels and Patterns of Exercise among Adults in Belgium. Int. J. Environ. Res. Public Health 2020, 17, 4144"

_ijerph, 2020, doi:10.3390/ijerph17165730_

Round 1

Reviewer 1 Report

Although Hew-Butler et al. make concise arguments against competitive sports during the pandemic, it should be noted that Constant et al. did not mention the specific exercise modalities used by people during pandemic. In addition, the sample is classified in "Highly active" or "less active", which could not be compared to sports athletes and competitors...

Since it is stated that "Modest exercise, performed with appropriate social distancing,face coverings, hand sanitizing, and disinfection strategies, should remain widely promoted by public health officials ",  Hew-Butler et al. could use Constant et al. findings in a positive manner to show the important difference between "active individuals exercising in safety" vs "sports athletes with high volume of training in unsafe enviroments".

Nevertheless, Hew-Butler et al. ignored the fact of how 55+ individuals demonstrated to exercise less during the pandemic, which should be noted as an increased risk factor for several diseases.

I consider important to "highlight the need for systematic, coordinated, and routine testing for all athletes to promptly identify asymptomatic athletes and remove them from play (i.e. interrupt virus transmission)." But I also consider that the authors should discuss how exercise IN A SAFE ENVIROMENT is in fact, a good medicine during the pandemic, as well as to discuss how exercising at home, and in safety (with appropriate profissional support, for example) should be recommended. This discussion should focus regular exercisers rather than sports athletes. While competitive sports safe return to training and competitions should be widely reinforced and debated, but not based solely on Constant et al., since they clearly stated several limitations.

Comments on published research should not be based solely on negative critiques. Constructive efforts could combine Constandt et al findings with Hew-Butler et al. considerations, ultimately promoting an idea of safe exercise promotion for every individual, including sports athletes. Unfortunately, Hew-Butler et al. focused only in the negative side of the limited findings from Constandt et al.

Therefore, I would recommend a more "constructive" critique in Hew-Butler et al. the introduction, using Constandt et al. paper to introduce their ideas, since it is highly important to promote exercise safely during the pandemic.

Hopefully, this discussion will lead to the promotion of exercise as an important health tool.
